# Classifying ball trajectories in invasion sports using dynamic time warping: A basketball case study

Yu Yi Yu[1], Paul Pao-Yen Wu[1]*, Kerrie Mengersen[1], Wade Hobbs[2,3]

1 Science and Engineering Faculty, Australian Research Centre, Centre of Excellence for Mathematical and Statistical frontiers (ACEMS), School of Mathematical Sciences, Queensland University of Technology, Brisbane, Australia, 2 Australian Institute of Sport, Bruce, Australia, 3 Faculty of Health Science, Exercise, Health & Performance, The University of Sydney, Sydney, Australia

* p.wu@qut.edu.au

**Data Availability Statement:** All relevant data are within the manuscript and its supporting information files.

**Funding:** This research was conducted by the Australian Research Council Centre of Excellence

## Abstract

Comparison and classification of ball trajectories can provide insight to support coaches and players in analysing their plays or opposition plays. This is challenging due to the innate variability and uncertainty of ball trajectories in space and time. We propose a framework based on Dynamic Time Warping (DTW) to cluster, compare and characterise trajectories in relation to play outcomes. Seventy-two international women's basketball games were analysed, where features such as ball trajectory, possession time and possession outcome were recorded. DTW was used to quantify the alignment-adjusted distance between three dimensional (two spatial, one temporal) trajectories. This distance, along with final location for the play (usually the shot), was then used to cluster trajectories. These clusters supported the conventional wisdom of higher scoring rates for fast breaks, but also identified other contextual factors affecting scoring rate, including bias towards one side of the court. In addition, some high scoring rate clusters were associated with greater mean change in the direction of ball movement, supporting the notion of entropy affecting effectiveness. Coaches and other end users could use such a framework to help make better use of their time by honing in on groups of effective or problematic plays for manual video analysis, for both their team and when scouting opponent teams and suggests new predictors for machine learning to analyse and predict trajectory-based sports.

## 1 Introduction

Basketball is a complex sport with many factors influencing a team's ability to score, such as strategy and tactics, team coordination, athletic abilities and the dynamics of the match during gameplay. Comparison and classification of ball trajectories can provide insight into plays run by a team or opposing teams to support coaches and players in preparation for, or adaptation during games. With the introduction of spatio-temporal tracking data of players and the ball, analyses attempting to better understand these influential factors have grown exponentially [1]. Automated clustering and classification of these data in relation to play outcomes can support coaches in analysing games, identifying trends and patterns over a large number of plays

for Mathematical and Statistical Frontiers (project number CE140100049) and funded in part by the Australian Government.

**Competing interests:** The authors report no conflict of interest.

and games, and highlighting particularly effective or ineffective plays for manual analysis (such as review of video footage).

While the analyses of strategy and tactics in basketball using spatio-temporal data are wide ranging, research attempting to group or cluster common play types is more limited [2]. Understanding the different types of ways in which a team or an opposition team moves the ball and their effectiveness could help coaches prioritise their focus when formulating offensive/defensive strategies. Previous research used K-means clustering and Euclidian distance to cluster the most common scoring plays from 380 professional soccer games [3]. The data included the location of all players on the field for the ten seconds prior to a goal being scored. Similarly, Hobbs et al used Euclidean distance along with hierarchical clustering to detect the 'playbook' of commonly used plays from English Premier League soccer data in order to quantify the value of transition in soccer [4]. Nistala and Guttag employed autoencoders, a type of unsupervised machine learning, again using Euclidian distance to cluster individual player trajectories in 3690 National Basketball Association basketball games [5]. A common challenge in these and similar studies is the quantification of the difference between two trajectories that might be of different lengths, with features in the trajectory that do not necessarily occur at exactly the same point in time. The cited papers commonly employ Euclidean distance. However, the Euclidean distance between two otherwise highly similar trajectories can be large if they are misaligned in time, especially in unsupervised situations where the true classification of a play is not necessarily known ahead of time.

Dynamic Time Warping (DTW) presents an approach to align features, such as a kick-out pass or a baseline drive, in space and time between trajectories of possibly different lengths. Aligning or registering features can be useful as two otherwise identical trajectories where one is slightly offset in space and/or time will have a large mathematical dissimilarity (e.g. Euclidean distance) even though they are similar in practice [6]. A registration method such as DTW can simultaneously align features and compute a dissimilarity measure which can then be used to better cluster trajectories. DTW has been used widely in a range of applications due to its computational efficiency and utility for feature alignment, such as the analysis of audio files [7] and in facial recognition as an accurate predictor for classification algorithms [8]. DTW classification has also been used to monitor production cycle rates for construction with up to 92% accuracy [9]. In addition, the DTW Barycentre Averaging (DBA) algorithm, which is based on DTW, can be used to generate an average trajectory for each cluster [10]. DTW has been used to for the purposes of curve registration when classifying events, such as the type of shot being made, using wearable sensor data or video data, with examples in tennis, squats and basketball [11–14]. There are comparatively fewer works in the analysis of basketball plays. [15] present an approach based on DTW registration of player trajectories and clustering of similar plays, representing similar plays with Gaussian mixture regression. However, their focus was on unsupervised clustering and they did not explicitly relate clusters to play outcomes. Another method that incorporated DTW was the extension of latent Dirichlet allocation to create a vocabulary of actions based on player tracking that combine to form a play Chen et al. (2015) [16]. This particular work largely ignores temporal effects and focuses on spatial characteristics of actions that make up a play.

In this paper, we propose a framework that relates play outcomes to automatically learned clusters of plays, vis a vis ball trajectories. We describe and illustrate the approach in the context of a substantive basketball case study with manually digitised spatio-temporal ball tracking data and play information including play outcomes (e.g. score, miss, turnover). By using the entire trajectory rather than just discrete features such as shot position and play duration, it is possible to better capture patterns emerging from interactions between players and the ball over time.

## 2 Materials and methods

In this paper, the following terms are utilised. Let a set of $N$ trajectories be described by $S_N = \{s_1, s_2, \ldots, s_N\}$. Each trajectory is composed of Cartesian coordinates and timestamp triplets, $(x, y, t)$, which are denoted for trajectory $n$ that is length $T$ as $s_n = \{s_{n1}, \ldots s_{nT}\}$. Let $a(i)$ be the average distance between the $i^{th}$ trajectory within cluster $c_k$ and all other trajectories in the same cluster and let $b(i)$ be the corresponding smallest average distance of the $i^{th}$ trajectory to all other points in the cluster.

Our framework comprises: (i) clustering of trajectory endpoints into areas (where a shot is taken or possession lost, Section 3.2), (ii) DTW to cluster trajectories terminating in each area (Section 3.1 and Section 3.2), and (iii) analysis of play outcomes for each cluster (Section 3.5). In addition, methods for validation (Section 3.3) and generating a representative trajectory from a cluster (Section 3.4) are also provided. This framework demonstrates how the combination of machine learning and statistical methods can be used in novel ways to address a practical problem.

### 2.1 Dynamic time warping

DTW is based on Levenshtein distance and finds an optimal coupling pathing between two sequences in a precomputed distance matrix [10]. In a general sense, the Levenshtein distance between two sequences is the minimum number of single-site edits (insertions, deletions, substitutions) required to change one sequence into another. Viewing two trajectories as a stream of data couplets, the distance matrix required for the Levenshtein distance is composed of distance values of the previous, current and next alignment of data pairs. As such, phase shifts between trajectories may be detected with the DTW algorithm which are not generally possible using Euclidean distance. This may be visualised in Fig 1.

Consider two trajectories $s_1$ and $s_2$. DTW achieves the optimal coupling by following a path of the smallest leading edge along a DTW distance matrix (also known as local cost matrix), $M$:

1. Initialise matrix of *size* (1: $length(s_1) \times 1$: $length(s_2)$)

2. For each Row, iterate through the columns and find the Euclidean distance of *item i* on the row element to *item j* on the column element. This is computed as a $n$ dimensional Euclidean distance.

3. Find the minimum Euclidean distance with neighbouring elements including itself; add the Euclidean distance calculated on the previous step and store in the respective location in the initialised matrix $M$.

4. Return matrix $M$.

The full DTW Algorithm is given in S3 –Appendices, Dynamic Time Warping Algorithm.

For each time point on a trajectory, the algorithm compares the distance to the same point on the second trajectory, and at the previous and following points (steps 2 and 3 above). The distance with the minimum distance is selected, where:

- $M[i-1][j]$: Known as a deletion, denotes $s_2$ is accelerated, where $s_1$ aligns with previous $s_2$ values.

- $M[i][j-1]$: Known as an insertion, denotes $s_2$ is decelerated, where $s_2$ aligns with previous $s_1$ values.

- $M[i-1][j-1]$: Known as a match, denotes $s_1$ aligns with $s_2$.

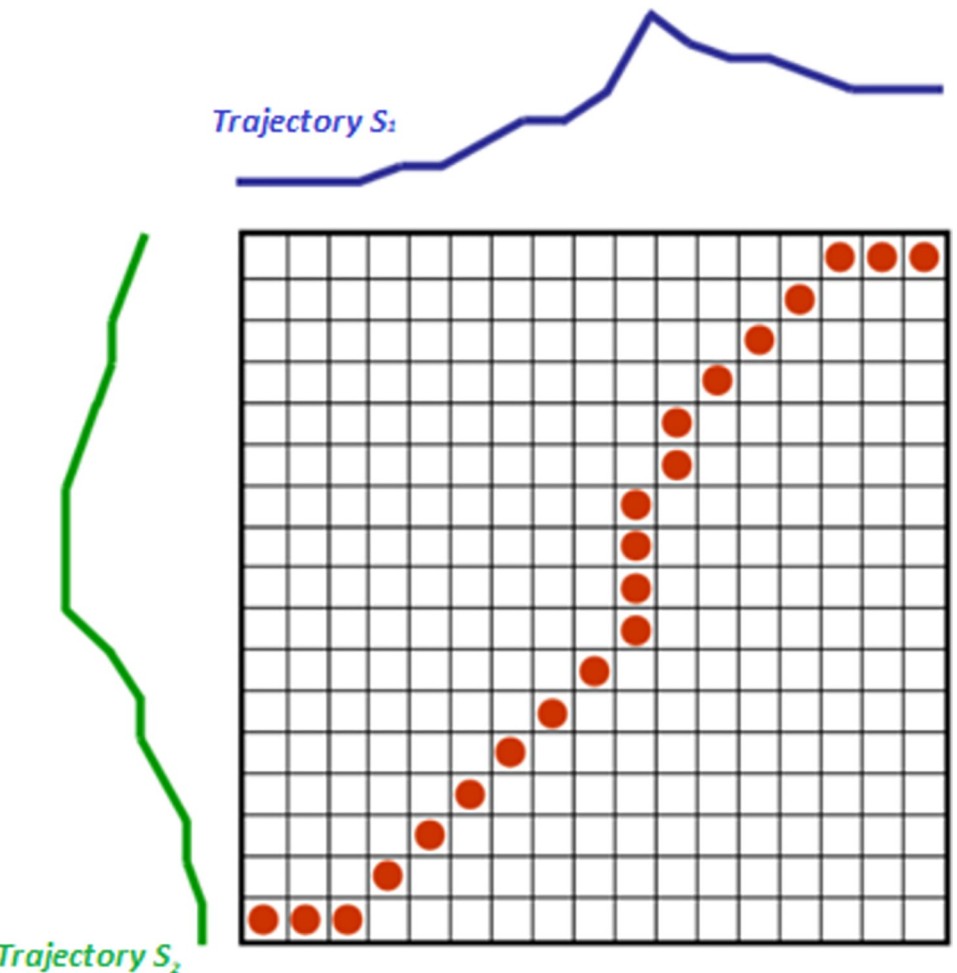

**Fig 1. Visualising DTW.** Two sample trajectories are analysed with the distance matrix provided. The optimal pathing may be found, represented by the red dots, which are alignment of the two trajectories (Anderberg, 1973).

Hence the DTW algorithm allows for "warping" of two trajectories to find a measure of minimum distance between them. We propose the application of DTW to three dimensional trajectories ($x,y,t$) to capture not just the pathway taken by the ball (dimensions in $x$ and $y$), but also the speed at which it moved along the trajectory (represented as time $t$). This is critical as the speed of execution of a play is just as important as the play (i.e. ball movements) itself in affecting scoring outcomes [17]. The full trajectory of a play was used to identify the variability and impact of temporal-based elements and was noted by industry partner the first 8 seconds of a play to be crucial in scoring [18].

In the case study below, we standardised each dimension prior to the computation steps of DTW. This differences in the range and variability amongst the $x,y$, and $t$ values; $t$ is bounded by the shot clock, and $x,y$ values bounded by the court dimension. To allow for greater stability and consistency when comparing across teams, the global standard deviation was used during the standardisation process.

The complexity of optimising the alignment of two ball movement trajectories increases exponentially with the number of time steps [18]. DTW provides one way to address this computational complexity by comparing discrete time points immediately before and after each point and by imposing the following conditions:

- Boundary condition: The path initialises from bottom left of matrix to top right. Mathematically: the $i$ and $j$ index initialises at (1,1) to ($X,Y$) (leading diagonal of matrix). This ensures that the analysis of trajectories is initialised correctly.

- Continuity condition: The path can only advance one step at a time. Mathematically: the $i$ and $j$ index can only increase by increments of 1. This ensures that each point within trajectories is matched in a sequential manner.

- Monotonic condition: The path may not trace back on itself. Mathematically: the $i$ and $j$ index cannot be decreased. The monotonicity condition mitigates potential infinite loops and mismatching in the temporal aspect. It also constrains the shape of trajectories, ensuring that the alignment follows through the leading diagonal of the distance matrix.

- Warping window condition: An optimal path is near the leading diagonal of the matrix.

## 2.2 Clustering methodology

Although DTW helps to find a measure of the closest match between every pair of trajectories, clustering is still a challenge. This arises due to the sheer diversity of ball movement trajectories with differing starting points (out of bounds, under the basket and so on), and unequal effect of parts of the trajectory on scoring outcomes. Furthermore, as temporal factors were included, trajectories may have a similar temporal path, but have very different spatial features. It was found that the endpoint of the trajectory from where the shot was taken was critically important from a scoring effectiveness and strategic perspective [19]. To address these challenges, we adopted a two-step approach. In the first step, we clustered trajectory endpoints using unsupervised K-means clustering. We refer to clusters of endpoints as areas in this paper. In the second step, for each area, we clustered using both spatial and temporal features from the data through hierarchical clustering of trajectories based on DTW distances to learn about the plays leading to endpoints in those areas.

Approaches such as K-means and mixture modelling are ideally suited to clustering endpoints as clusters are formed around a centroid which has a practical spatial interpretation on the court. Here we chose K-means for its simplicity and efficiency for single point classification [20–22]. For step (ii), we chose hierarchical clustering to minimise the variance within the clusters derived from pairwise DTW distance measures. The "ward.d2" method [23] in the "hclust" package in R (Anderberg, 1973) was used, which begins with every trajectory in its own cluster, and pairs of clusters are incrementally combined in order of similarity (i.e. least distance). This agglomerative process produces a tree where in which the distance (i.e. height) to the leaves represents a threshold for similarity within clusters; the tree can be pruned at arbitrary thresholds ranging from every trajectory in its own cluster to all trajectories in one cluster. As the clustering task is unsupervised, i.e. there is no ground truth of how trajectories should be assigned to clusters, we used validation indices and specifically the Silhouette index.

Note that this dataset contained a number of trajectories composed of only a single point. These were found to be an artefact of the trajectory encoding process in situations such as an offensive rebound, or a violation occurred prior to the ball moving. These encoding artefacts were filtered out and removed prior to analysis.

## 2.3 Validation indices

A key challenge of unsupervised clustering is the selection of the number of clusters, $K$. Given the volume and diversity of plays over multiple games, it is infeasible to manually annotate trajectories to form a ground truth for clustering. We adopted the Silhouette Index (SI) to find an appropriate value of $K$ because it is a normalised metric balancing similarity within a cluster

and dissimilarity between clusters [24]. Other indices are available and were also evaluated (See S3 –Appendices, Validation indices and Results) [25]. The SI, defined in Eq 2, is based on cohesion (distance of data relative to own cluster centre) and separation (distance of data relative to other cluster centres); maximal cohesion to separation occurs when the index is 1, suggesting a clustering configuration where the trajectories are well matched to their respective clusters.

$$s(i) = \frac{b(i) - a(i)}{\max\{a(i), b(i)\}}$$

The number of clusters can be chosen by plotting the SI index against the number of clusters and finding the 'elbow' [22] as this helps to achieve a near-maximal SI index. This approach was applied to both the choice of the number of clusters for endpoint clustering and hierarchical clustering of trajectories.

## 2.4 Dynamic time warping barycentre averaging

To represent each cluster, a representative trajectory was computed using the DTW Barycentre Averaging DBA algorithm [10]. This algorithm aims to minimise the sum of DTW distances computed between the average trajectory to trajectory in the cluster. Assume that $N$ trajectories are grouped in $K$ clusters. The set of clusters is denoted by $C_k = \{c_1, c_2, \ldots c_K\}$. The $k^{th}$ cluster is represented as a list of trajectories $c_k = \{c_1^k, c_2^k, \ldots, c_{U_k}^k\}$, where $U_k$ is the number of trajectories in the cluster. The centroid $\bar{c}_k$ of cluster $c_k$ is calculated as the Barycentre average of the component trajectories (1) with the following algorithm:

1. Initialise an arbitrary trajectory as the centroid $\bar{c}_k$.

2. Iterate through the trajectories $c_i^k$ inside the chosen cluster $c_k$.

   a. Compute the DTW aligned trajectories $c_i^k$ between the centroid and each trajectory $i$ in a cluster.

   b. Update the centroid based on the barycentre of the coordinates iteratively, as defined in Eq 1.

$$\bar{c}_k = \frac{c_1^k + c_2^k + \cdots + c_{U_k}^k}{U_k}$$

(2)

## 2.5 Play outcomes

We used a number of metrics to characterise the outcomes of a play (i.e. possession). These included the mean and standard deviation in: play duration, distance travelled, heading angle, and change in heading angle. Note that the heading angle is measured relative to the horizontal line joining the two baskets, and is computed for each segment (pair of two consecutive sample points $j$ and $j+1$) of a trajectory.

$$Heading\ Angle = \arctan\left(\frac{y_{j+1} - y_j}{x_{j+1} - x_j}\right)$$

(3)

In addition, the scoring rate of trajectory clusters was used as a measure of scoring outcome. This is the ratio of scored shots to total number of shots attempted (3).

$$Scoring\ Rate = \frac{no.Scored\ Shots}{Total\ no.Shots}$$

## 2.6 Case study

Existing data from 72 women's international basketball games ranging from the 2014 FIBA World Championships to the 2016 Rio Olympic Games were analysed [26]. The data consist of manually tracked *x* and *y* coordinates with a frequency between 1~6 Hz, and metadata of event sequences manually encoded whilst the coordinates were tracked. Metadata includes events such as turnovers, fouls, violations, shots and shot outcomes.

In this paper, we focused on a case study on three teams with the greatest number of games: Australia (11 games), USA (12 games), and Japan (12 games).

# 3 Case study results

Using the elbow method in conjunction with having more precise trajectories, we selected seven clusters for clustering trajectory endpoints using K-means, which coincided with tactically interpretable areas for shot analysis in basketball (Fig 2). The Silhouette Plot can be seen in Figs 10 and 11 S3 Appendices.

The elbow method was also used to help guide the choice of a threshold for pruning the tree obtained from hierarchical clustering of trajectories. As trajectories were first clustered by endpoint using K-means into areas, and then again using DTW and hierarchical clustering, there were at times differing numbers of trajectory clusters per area. Trajectory clusters are recorded in S5, S9 and S13. In addition to the elbow method, the height for pruning was also informed subjectively in terms of manual interpretation. This comprised maintaining similar sizes of clusters within an area, and minimal variation in trajectory spatial length and temporal length within each cluster. The distance for pruning was chosen based on visual inspections where preliminary screening of trajectory grouping was conducted. Visual cues included having equally distributed clusters with minimal variations in parameters within clusters, both spatially and temporally (Fig 3).

The clusters (S6, S9 and S11 Appendices) were then ordered by the most common play to the least common play per area, and summary statistics including spatial distance, temporal duration, scoring rate were computed (Tables 3, 4 and 5 in S2 Appendix).

Consider for instance scoring under the basket, area three, for Australia (Table 1). In addition to the number of trajectories in each cluster and their relative frequency, the mean and

## AUS: Kmeans Division

**Fig 2. Kmeans for Australia dataset.**

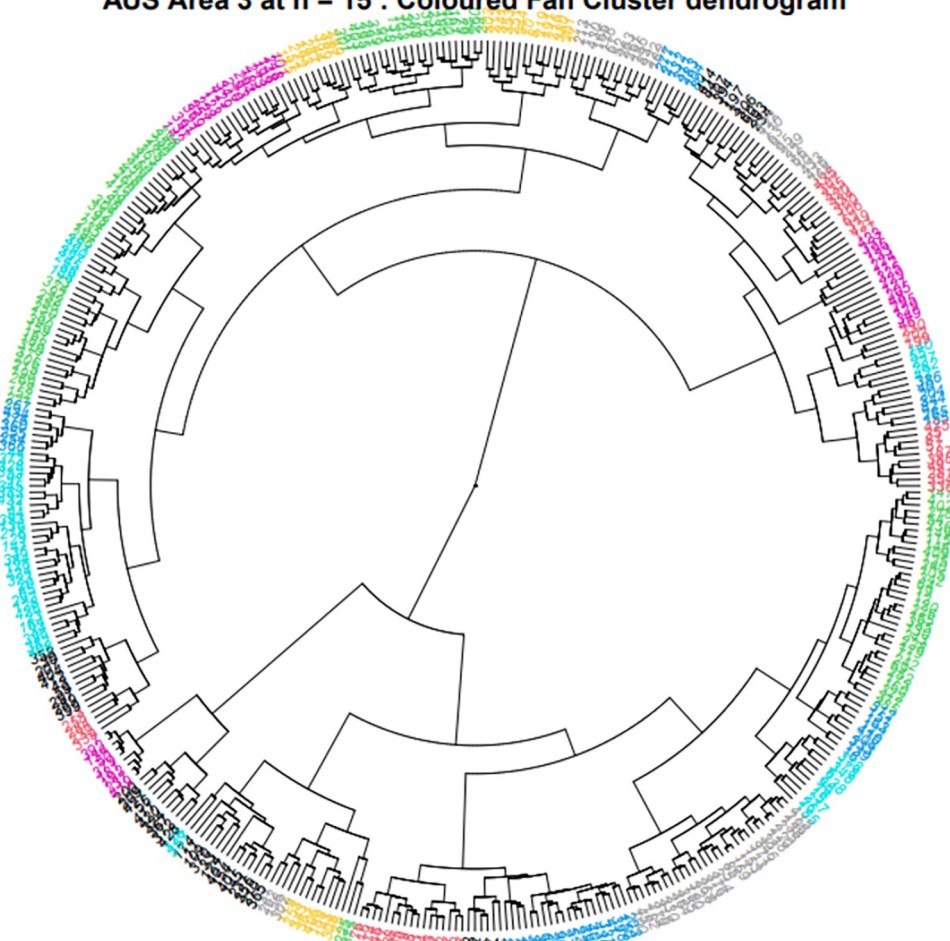

**Fig 3. Fan dendrogram of cluster results for Australia Dataset Area 3.** Each cluster is represented by a colour.

standard deviation (SD) in duration (in seconds), the mean and sd change in heading (degrees), the current heading (degrees), and scoring rate is recorded. Here, the heading is the angle between the ball's current position and the basket. The change in heading is the difference between the current heading and the previous heading. These parameters are measured in degrees ranging between -180 degrees to 180 degrees from the trajectorial point in respect to the goal post, with a positive score. The approach helps to highlight that clusters 16 in this area correspond to particularly high and particularly low scoring rates, respectively. Inspection of the trajectory (Fig 4) provides additional visual aid in understanding the play and context of ball movement which may assist in understanding the effects of spatial temporal factors to scoring rate.

Speed in trajectories may also be visualised to add an additional dimension for interpretation for the trajectorial plots. This can be achieved by colouring the trajectorial based on possession time. An example is given in Figs 5 and 6, where two temporal trajectorial plots are given for Australian trajectories for Area 3 Cluster 16 and Area 3 Cluster 14.

Clustering results can be compared between countries to help highlight potential trends, strengths and weaknesses in a team relative to the competition. In particular, we focused on results from the summary data sorted on parameters such as scoring rate and possession time,

**Table 1. Sample summary statistics for Area 3 Australia data.**

| Area | Cluster | Trajectories | Frequency of Execution (%) | Mean Trajectory Duration (sec) | SD Trajectory Duration (sec) | Mean Change in Heading (deg) | SD Change in Heading (deg) | Mean Heading (deg) | SD Heading (deg) | Scoring Rate (%) | Mean distance travelled (Foot) | SD distance travelled (Foot) |
|---|---|---|---|---|---|---|---|---|---|---|---|---|
| 3 | 1 | 34 | 3.6403 | 5.5882 | 1.7773 | 1.971 | 8.761 | -3.667 | 18.621 | 44.118 | 60.8751 | 9.2206 |
| 3 | 2 | 31 | 3.3191 | 8 | 2.1602 | -0.453 | 5.581 | -11.333 | 13.591 | 29.032 | 74.5008 | 7.97 |
| 3 | 3 | 30 | 3.212 | 3.8333 | 1.3153 | 2.733 | 11.832 | 13.401 | 23.64 | 60 | 32.6436 | 10.9074 |
| 3 | 4 | 22 | 2.3555 | 12.0455 | 1.3965 | 2.361 | 3.289 | 8.938 | 8.417 | 27.273 | 74.4336 | 10.839 |
| 3 | 5 | 21 | 2.2484 | 12.0952 | 1.9724 | -0.682 | 4.28 | 2.613 | 9.918 | 57.143 | 74.0438 | 7.5899 |
| 3 | 6 | 20 | 2.1413 | 8.4 | 1.875 | 1.73 | 4.922 | 20.139 | 9.987 | 35 | 71.1293 | 7.1378 |
| 3 | 7 | 20 | 2.1413 | 14.75 | 2.4682 | 3.455 | 1.982 | 14.158 | 5.999 | 35 | 77.765 | 6.5939 |
| 3 | 8 | 20 | 2.1413 | 11.8 | 2.2618 | -2.326 | 5.26 | -12.347 | 9.958 | 30 | 72.6537 | 9.6632 |
| 3 | 9 | 17 | 1.8201 | 6.4706 | 1.2307 | 2.687 | 6.142 | 19.211 | 14.461 | 23.529 | 71.1083 | 6.2399 |
| 3 | 10 | 14 | 1.4989 | 15.3571 | 2.17 | 2.699 | 3.976 | 8.239 | 11.826 | 28.571 | 88.4606 | 11.3146 |
| 3 | 11 | 14 | 1.4989 | 10.3571 | 2.4371 | 0.693 | 6.406 | -2.699 | 14.536 | 42.857 | 41.3235 | 13.3192 |
| 3 | 12 | 13 | 1.3919 | 11.7692 | 2.4547 | 4.927 | 3.913 | 19.899 | 8.497 | 30.769 | 71.7065 | 15.6306 |
| 3 | 13 | 12 | 1.2848 | 16.4167 | 2.3533 | 0.922 | 2.366 | 13.253 | 9.672 | 33.333 | 82.3777 | 11.2804 |
| 3 | 14 | 11 | 1.1777 | 19.5455 | 3.1101 | 0.745 | 2.447 | 5.97 | 9.757 | 45.455 | 92.1455 | 12.9826 |
| 3 | 15 | 11 | 1.1777 | 8.4545 | 1.8635 | 1.392 | 6.984 | 6.303 | 21.497 | 36.364 | 46.5183 | 14.85 |
| 3 | 16 | 11 | 1.1777 | 15.1818 | 2.3587 | -0.172 | 3.449 | 11.837 | 9.557 | 72.727 | 91.1302 | 11.0612 |
| 3 | 17 | 10 | 1.0707 | 10.8 | 1.3984 | 1.014 | 5.134 | -19.727 | 6.52 | 30 | 78.8633 | 10.157 |
| 3 | 18 | 10 | 1.0707 | 18.1 | 2.4698 | 3.254 | 0.848 | 6.434 | 8.663 | 30 | 83.087 | 7.6823 |
| 3 | 19 | 10 | 1.0707 | 8.2 | 1.3984 | 0.596 | 9.969 | 1.57 | 34.183 | 50 | 28.939 | 6.3694 |
| 3 | 20 | 9 | 0.9636 | 3.1111 | 0.928 | -6.142 | 28.831 | 8.629 | 38.858 | 33.333 | 8.3526 | 6.401 |
| 3 | 21 | 9 | 0.9636 | 5.3333 | 1.9365 | 3.06 | 6.136 | -9.448 | 20.162 | 44.444 | 64.2681 | 11.8536 |
| 3 | 22 | 9 | 0.9636 | 14.3333 | 1.7321 | 1.833 | 2.693 | -3.146 | 9.941 | 33.333 | 77.7904 | 9.2834 |
| 3 | 23 | 8 | 0.8565 | 22.75 | 2.3755 | -1.197 | 1.667 | -4.314 | 8.365 | 37.5 | 95.5013 | 22.0408 |
| 3 | 24 | 8 | 0.8565 | 13.625 | 2.3261 | 0.779 | 5.053 | 1.266 | 15.533 | 0 | 60.368 | 16.4986 |
| 3 | 25 | 7 | 0.7495 | 19.2857 | 1.8898 | 0.625 | 3.249 | -4.927 | 5.46 | 14.286 | 89.2002 | 14.9324 |
| 3 | 26 | 7 | 0.7495 | 11.4286 | 1.5119 | 2.739 | 5.243 | -1.003 | 16.175 | 14.286 | 63.8742 | 16.2557 |

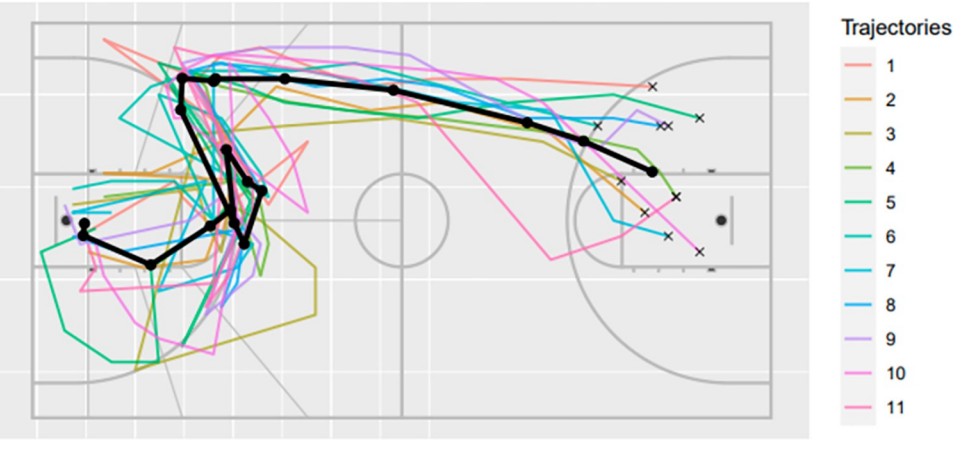

**Fig 4. Trajectorial plots for Australia Area 3 Cluster 16.**

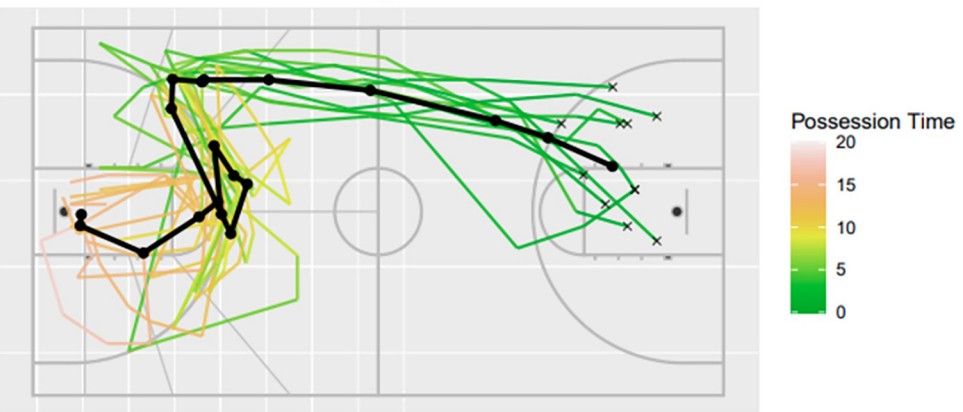

**Fig 5. Temporal Trajectorial plots for Australia Area 3 Cluster 16.**

to identify key trends amongst plays that have similar properties. We manually identified a trajectory from the USA (Fig 7) that somewhat resembles Australia's trajectory of Area 3 Cluster 16 for comparison.

## 4 Discussion

Fundamentally, the proposed framework allows us to identify, interrogate and compare clusters of similar ball movement trajectories; these trajectories present one way to represent basketball plays. We discuss some ways in which this analysis can be used to analyse play outcomes via the scoring rate of different clusters (Section 5.1), provide insight into spatial and temporal tendencies and/or patterns (Section 5.2 and 5.3, respectively), and make comparisons between teams (Section 0). End users such as coaches and athletes may sort and filter clusters (i.e. plays) based on modelled factors, including the scoring rate, distance travelled, possession time and change of heading (as an indicator of variability). This helps to facilitate efficient and targeted review and analysis of plays (such as of video footage) to support training and game preparation. We discuss below a number of different perspectives for analysis.

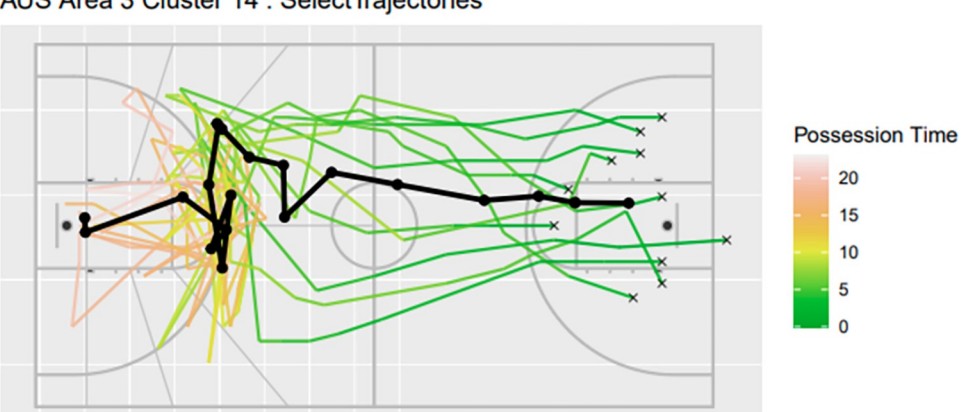

**Fig 6. Temporal Trajectorial plots for Australia Area 3 Cluster 14.**

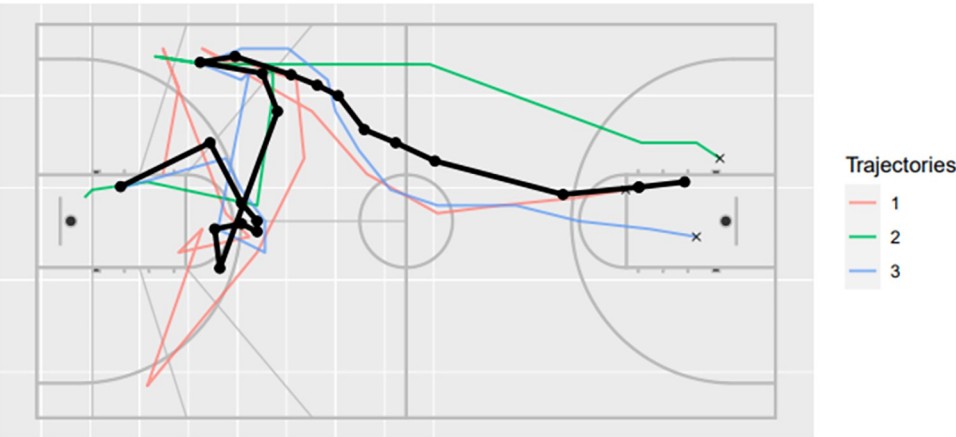

**Fig 7. Trajectorial plots for USA Area 5 Cluster 27.**

## 4.1 Scoring rate of clusters

One approach for analysing successful plays is to compare trajectory clusters with high scoring rates and which terminate in the same area of the court. For instance, Table 1 shows that both Area 3 Cluster 3 and Area 3 Cluster 16 for Australia have high scoring rates of 60.0% and 72.7%, respectively, but that arises from very different types of play. The mean duration for cluster 3 is 3.8 seconds, compared to 15.2 seconds for cluster 16. The former corresponds to fast breaks, which are known to typically result in a higher scoring rate [27,28].

However, our trajectory clustering also reveals that cluster 3 has lower mean heading changes compared to cluster 16, 74 degrees compared to 304 degrees, respectively (Table 1). Potentially, changes in direction, either more frequent and/or larger in magnitude, could be contributing towards the higher scoring rate. We then sorted Table 1 by mean heading change to identify the top ten and bottom ten clusters in terms of heading change. The scoring rate for the top ten was 29%, on average, compared to just 14%, on average, for the bottom ten. This builds upon findings from previous work that found entropy was a contributing factor to play effectiveness [26]. Based on the plays in cluster 3 and 16, end users could review video footage to support game preparation in terms of plays and their execution.

## 4.2 Spatial tendencies

High level summary analysis may help to understand the overall playstyle and score rate for a country. Here we look at patterns from a spatial context.

In Table 1, we found that there were 534 plays by Australia that had a right-hand bias where much of the play occurred with the ball in the right half of the court, vis a vis the mean heading angle was positive. In contrast, 400 plays had left-hand bias (negative mean heading angle). However, clusters with right-hand court bias had, on average, a 30% scoring rate compared to 25% for left-hand court biased clusters. This type of analysis could be helpful for training of the home team, or for preparing against an opponent team.

## 4.3 Temporal context

By sorting the summary table by possession time, coaches and players can view clusters that perform well during fast breaks compared to long possession time. The fastest ten clusters have an average scoring rate of 30% whilst the slowest ten clusters have an average of 19%,

which may indicate fast plays have better scoring rates. However, upon closer inspection, the cluster size in these extremities (both slowest and fastest trajectories) have low occurrence rates (proportion of trajectories). Thus, it is important to further analyse these trajectories to support decision making. For instance, trajectory plots, such as those used in Figs 5 and 6 may provide high level insight on both spatial temporal aspects of a play (i.e. court location and speed of movement). However, review of video footage is likely also necessary; the proposed model helps to guide end users to identify which plays to review in light of time pressure and/ or limited time.

There can be considerable variation in the scoring rate of fast break plays terminating in the same area. Consider for instance cluster 20 of Area 3 with a scoring rate of 33% versus cluster 3 of Area 3 with 60% (Table 1). The former accounts for a greater proportion of plays (3.2%) compared to the latter (0.96%). Although the possession time is quite similar (around 3 seconds of play) for both, further investigation, such as of spatial patterns of the trajectories and review of video footage could help identify the reason for this disparity. An example of a high-level spatial pattern is that out of the top ten fastest plays, the scoring rate was 41% on average for plays with a right-hand court bias (i.e. positive mean heading angle), compared to 13% for those with left-hand court bias.

## 4.4 Comparisons between teams

Comparisons between teams can also offer insight to support preparation for competition. The equivalent trajectory cluster for Australia's Area 3 Cluster 16 was Area 5 Cluster 27 for team USA based on endpoint area and trajectory plot. Factors such as the frequency of execution, possession time, mean heading and scoring rate can be used identify potential differences in the play and its execution between the two teams. In this example, cluster 27 for team USA was run relatively rarely (0.31% of the time) and with a significantly lower scoring rate (0%) than cluster 16 for Australia (1.18% proportion of plays with 72.73% scoring rate). A key difference appears to be the speed of execution, which is slower for team USA (19s) compared to Australia (15.81s).

## 4.5 Conclusion

Our DTW based framework enables analysis and interpretation of complex trajectories over many games and teams, and the generation of key summary metrics of play outcomes. Comparison of trajectory clusters within and between countries can assist in identifying key patterns to support coaching and preparation for games. A key component of this is the ability to highlight key plays for targeted video review from a large database of game footage, assisting end users with limited time resources. A common methodology used in current and previous research utilises Euclidean distances [3–5] which does not consider data alignment and will miss out similar plays when analysed under unsupervised conditions. Existing works that used DTW in basketball tended to focus on player movements, and did not explicitly relate trajectories to play outcomes with a focus on both spatial and temporal characteristic of the trajectories [15,16]. Our framework contributes a practical means to cluster spatio-temporal trajectories and relate them to play outcomes, using ball movement data. However, the use of k-means to cluster trajectory endpoints is limited in that it cannot take into account boundary conditions associated with court boundaries. In addition, there is immense diversity in trajectories and determining how many clusters to use is somewhat subjective; potentially, a combination of using a vocabulary of discrete actions [16] and a Diricihlet Process Mixture Model [29] could better cluster trajectories and inherently finds the optimal number of clusters.

This paper illustrated some insights that could be obtained through DTW clustering, including support for conventional wisdom of higher scoring rates with fast breaks, and clusters where more changes in direction (higher mean heading change) in the play was also associated with high scoring rates. Interestingly, the top ten fast break clusters for team Australia revealed a significant disparity between those that had a bias towards the right-hand side of the court, which were more effective than those with a left-hand side bias. In addition to these summary metrics, trajectory plots could also be used to capture spatial layout of the trajectory and its speed of execution. This provides an opportunity for coaches and players to understand the effects of spatio-temporal factors into plays on a macro level, in addition to the micro level of video review.

Potential future work includes optimisation of the clustering process to better account for large numbers of clusters with a small number of trajectories. The current framework utilises a basic implementation of both DTW and k-means clustering; more sophisticated approaches, such as PrunedDTW, SparseDTW, FastDTW, and MultiscaleDTW, could potentially enable faster computation and better clustering results [30]. Alternative techniques for clustering and even mixture modelling could be investigated, especially if integrating clustering with trajectory alignment. A higher resolution dataset with better sampling consistency may also assist in the DTW and clustering process; further investigation could include the breakdown of trajectories into discrete actions as part of a vocabulary of basketball actions [16]. Finally, extension of the work to include both ball and player trajectories could provide further detail and insight to support coaches and athletes.

## Supporting information

**S1 Appendix. Computer software.** Dynamic time warping algorithm. Dynamic time warping barycentre averaging. Validation indices and results. Silhouette index plot for K-means. Silhouette index plot for K-means (Australia Area 3).
(DOCX)

**S2 Appendix.** Table 3. Australia Summary Statistics. Table 4. Japan Summary Statistics. Table 5. USA Summary Statistics.
(DOCX)

**S3 Appendix. Australia dendrograms.**
(PDF)

**S4 Appendix. Australia trajectories by cluster.**
(PDF)

**S5 Appendix. Australia trajectories by cluster coloured by time.**
(PDF)

**S6 Appendix. Japan dendrograms.**
(PDF)

**S7 Appendix. Japan trajectories by cluster.**
(PDF)

**S8 Appendix. Japan trajectories by cluster coloured by time.**
(PDF)

**S9 Appendix. USA dendrograms.**
(PDF)

**S10 Appendix. USA trajectories by cluster.**
(PDF)

**S11 Appendix. Australia trajectories by cluster coloured by time.**
(PDF)

**S1 Data.**
(ZIP)

## Author Contributions

**Conceptualization:** Paul Pao-Yen Wu, Wade Hobbs.

**Data curation:** Wade Hobbs.

**Formal analysis:** Yu Yi Yu.

**Funding acquisition:** Paul Pao-Yen Wu, Kerrie Mengersen.

**Investigation:** Yu Yi Yu, Paul Pao-Yen Wu.

**Methodology:** Yu Yi Yu, Paul Pao-Yen Wu.

**Resources:** Wade Hobbs.

**Supervision:** Paul Pao-Yen Wu, Kerrie Mengersen.

**Visualization:** Yu Yi Yu.

**Writing – original draft:** Yu Yi Yu.

**Writing – review & editing:** Yu Yi Yu, Paul Pao-Yen Wu, Kerrie Mengersen, Wade Hobbs.

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
