## [Decision Letter · Decision Letter 0]

10 Jan 2022

PONE-D-21-32152Classifying ball trajectories in invasion sports using Dynamic Time Warping: a basketball case studyPLOS ONE

Dear Dr. Wu,

Thank you for submitting your manuscript to PLOS ONE. After careful consideration, we feel that it has merit but does not fully meet PLOS ONE’s publication criteria as it currently stands. Therefore, we invite you to submit a revised version of the manuscript that addresses the points raised during the review process.

Although the manuscript was generally well received by the reviewers, its contributions should be made more explicit and the methodology should be better justified and explained.  Please ensure that your decision is justified on PLOS ONE’s publication criteria and not, for example, on novelty or perceived impact.

We look forward to receiving your revised manuscript.

Kind regards,

Jean-Christophe Nebel, Ph.D

Academic Editor

PLOS ONE

Journal Requirements:

2. In your Methods section, please provide additional information regarding the permits you obtained for the work. Please ensure you have included the full name of the authority that approved the field site access and, if no permits were required, a brief statement explaining why

Reviewers' comments:

Reviewer's Responses to Questions

**Comments to the Author**

1. Is the manuscript technically sound, and do the data support the conclusions?

Reviewer #1: Partly

Reviewer #2: Yes

2. Has the statistical analysis been performed appropriately and rigorously? 

Reviewer #1: No

Reviewer #2: Yes

3. Have the authors made all data underlying the findings in their manuscript fully available?

Reviewer #1: Yes

Reviewer #2: Yes

4. Is the manuscript presented in an intelligible fashion and written in standard English?

Reviewer #1: Yes

Reviewer #2: Yes

5. Review Comments to the Author

Reviewer #1: Review on Manuscript: “Classifying ball trajectories in invasion sports using Dynamic Time Warping: a basketball case study”

NOTE: Please see the PDF file, I included some links and recommendation for further study.

In this manuscript, the authors presented a method to compare and to classify ball trajectories based on Dynamic Time Warping (DTW). The paper is well-written with some arguments on how DTW solved problem of measuring distance between two trajectories that were misaligned in time. The authors argued that DTW is useful for feature alignment in space and time. The idea of this research is interesting, however I am not really convinced by how the authors explained the details of the proposed method and how the proposed method contributes in the field of time series analysis based on machine learning.

Several comments regarding this paper:

1. What is the main contribution of this paper? The authors should explain contribution of the manuscript in the relevant field. In fact, Dynamic Time Warping (DTW) and K-Means clustering have been used widely in time series analysis across different research fields. In terms of time series analysis, DTW and K-Means clustering are not new methods.

2. Dynamic Time Warping is good for similarity measure between two time series data. However, this algorithm has a time and space complexity of O(Sn1, Sn2) where Sn1 and Sn2 are the lengths of the respective time series sequences between which DTW distance is to be calculated (see line 84-89). On the other hand, there are several other implementations of DTW that are faster in implementation, such as PrunedDTW , SparseDTW , FastDTW , and MultiscaleDTW . Why did the authors choose original DTW to be used in this research instead of considering other faster version of DTW?

3. What is the size of the local cost Matrix M?

4. In addition to the chosen DTW method, the authors also choose K-Means clustering by justifying that K-Means is simple and efficient (line 159). However, K-Means clustering is well-known as a “lazy classifier” and it takes more computational time in classification process. Simplicity in implementation does not mean that this algorithm is appropriate to solve the problem. More justification is needed and please compare the proposed classification method with other classification method.

5. In line 143-144, the authors wrote: “parameters such as boundary condition, continuity condition, monotonic condition and warping window condition were left the same during the analysis.” More detailed explanation on these parameters are needed to make this research reproducible.

6. In this research, the authors used a dataset of 72 women’s international basketball games ranging from the 2014 FIB world Championship to the 2016 Rio Olympic Games. Was there any specific justification in choosing this dataset? In fact, there are other datasets available in the internet, such as a dataset provided by Rajiv Shah.

7. More detailed discussions are needed, for instance:

a. How does this research contribute in the field of machine learning, especially compared with previous works of time series analysis in sports data? Comparison with previous works is needed to see that the proposed method is better than current state-of-the-arts in the field.

b. What are limitations of this research? What is the trade-off of implementing DTW and K-Means clustering?

*** EOF ***

Reviewer #2: I liked how this paper took a slightly different data science approach and applied it to get some useful sports analytic insights.

It's hard for me to believe that this is the first application of DTW to sports trajectories. Also, citing some other direct work like this would be good: http://www.lukebornn.com/papers/miller_ssac_2017.pdf So just ensuring that previous related work is acknowledged would be valuable. But even if DTW has been done before for trajectories, this paper did focus on womens basketball games.

I appreciate the authors taking time to visualize DTW and help everyone understand how the technique works. I like the detailed background and the application to the games. I could follow how DTW work and the resulting insights. Really great job sharing lots of detail.

For the plays, could you explain why you focused on the full court plays? I have seen previous work that would just focus on trajectories after the ball passed half court. No need to change the analysis, just explaining your decision would be useful.

I like that the paper doesn't try to extend itself too far, but is a solid piece of work on applying DTW to basketball.

6. PLOS authors have the option to publish the peer review history of their article (what does this mean?). If published, this will include your full peer review and any attached files.

Reviewer #1: No

Reviewer #2: **Yes: **Rajiv Shah

---

## [Author Response · Author response to Decision Letter 0]

3 Jun 2022

Dear editor

We would like to sincerely thank the reviewers for their comments in helping us to improve our paper. Please refer to our attached response to reviewers where our responses below are highlighted in yellow, and excerpts from changes made in the paper are highlighted in gray.

Thanks

Alan

Review on Manuscript: “Classifying ball trajectories in invasion sports using Dynamic Time Warping: a basketball case study”.

In this manuscript, the authors presented a method to compare and to classify ball trajectories based on Dynamic Time Warping (DTW). The paper is well-written with some arguments on how DTW solved problem of measuring distance between two trajectories that were misaligned in time. The authors argued that DTW is useful for feature alignment in space and time. The idea of this research is interesting, however I am not really convinced by how the authors explained the details of the proposed method and how the proposed method contributes in the field of time series analysis based on machine learning.

Several comments regarding this paper:

 What is the main contribution of this paper? The authors should explain contribution of the manuscript in the relevant field. In fact, Dynamic Time Warping (DTW) and K- Means clustering have been used widely in time series analysis across different research fields. In terms of time series analysis, DTW and K-Means clustering are not new methods.

Our work was motivated primarily to provide decision support, via the Australian Institute of Sport, to the national women’s Olympic basketball team. Hence, our paper focuses on the combination of relevant and accessible methods to generate new insights about plays and their relation to play outcomes. Thus, our contribution is the development of a framework, comprising existing methods, to relate clusters of plays, as represented by ball movement trajectories, to play outcomes. 

We have clarified this in the abstract, introduction, methods and discussion/conclusion: 

Abstract: 

(line 21) We propose a framework based on Dynamic Time Warping (DTW) to cluster, compare and characterise trajectories in relation to play outcomes.

Introduction: 

(line 90) In this paper, we propose a framework that relates play outcomes to automatically learned clusters of plays, vis a vis ball trajectories. We describe and illustrate the approach in the context of a substantive basketball case study…

Methods:

(line 105) Our framework comprises: (i) clustering of trajectory endpoints into areas (where a shot is taken or possession lost, Section 3.2), (ii) DTW to cluster trajectories terminating in each area (Section 3.1 and Section 3.2), and (iii) analysis of play outcomes for each cluster (Section 3.5). In addition, methods for validation (Section 3.3) and generating a representative trajectory from a cluster (Section 3.4) are also provided. This framework demonstrates how the combination of machine learning and statistical methods can be used in novel ways to address a practical problem. 

Discussion:

(line 359) Our DTW based framework enables analysis and interpretation of complex trajectories over many games and teams, and the generation of key summary metrics of play outcomes

(line 385) Potential future work includes optimisation of the clustering process to better account for large numbers of clusters with a small number of trajectories. The current framework utilises a basic implementation of both DTW and k-means clustering; more sophisticated approaches, such as PrunedDTW, SparseDTW, FastDTW, and MultiscaleDTW, could potentially enable faster computation and better clustering results (31). Alternative techniques for clustering and even mixture modelling could be investigated, especially if integrating clustering with trajectory alignment. A higher resolution dataset with better sampling consistency may also assist in the DTW and clustering process; further investigation could include the breakdown of trajectories into discrete actions as part of a vocabulary of basketball actions (15). Finally, extension of the work to include both ball and player trajectories could provide further detail and insight to support coaches and athletes. 

 Dynamic Time Warping is good for similarity measure between two time series data. However, this algorithm has a time and space complexity of O(Sn1, Sn2) where Sn1 and Sn2 are the lengths of the respective time series sequences between which DTW distance is to be calculated (see line 84-89). On the other hand, there are several other implementations of DTW that are faster in implementation, such as PrunedDTW1, SparseDTW2, FastDTW3, and MultiscaleDTW4. Why did the authors choose original DTW to be used in this research instead of considering other faster version of DTW?

Given the applied focus of the work, one of our goals was for ease of comprehension in our framework and thus trust and uptake by potential users including sports scientists, coaches and other stakeholders. In addition, the task of relating ball movement trajectories to play outcomes was not computationally intensive and could easily be addressed by DTW. However, we agree that future extension of this work needs to consider other DTW methods, which might be faster and/or potentially better clustering outcomes. To this end, we have included the following in the Discussion:

(line 385) Potential future work includes optimisation of the clustering process to better account for large numbers of clusters with a small number of trajectories. The current framework utilises a basic implementation of both DTW and k-means clustering; more sophisticated approaches, such as PrunedDTW, SparseDTW, FastDTW, and MultiscaleDTW, could potentially enable faster computation and better clustering results (31). Alternative techniques for clustering and even mixture modelling could be investigated, especially if integrating clustering with trajectory alignment. A higher resolution dataset with better sampling consistency may also assist in the DTW and clustering process; further investigation could include the breakdown of trajectories into discrete actions as part of a vocabulary of basketball actions (15). Finally, extension of the work to include both ball and player trajectories could provide further detail and insight to support coaches and athletes. 

 What is the size of the local cost Matrix M?

It is T_1 by T_2, which are the lengths of two trajectories s_1 and s_2 being aligned as per the Methods:

(line 121) Consider two trajectories s_1 and s_2. DTW achieves the optimal coupling by following a path of the smallest leading edge along a DTW distance matrix (also known as local cost matrix), M: 

 Initialise matrix of size(1:length(s_1 )×1:length(s_2 )) 

 In addition to the chosen DTW method, the authors also choose K-Means clustering by justifying that K-Means is simple and efficient (line 159). However, K-Means clustering is well-known as a “lazy classifier” and it takes more computational time in classification process. Simplicity in implementation does not mean that this algorithm is appropriate to solve the problem. More justification is needed and please compare the proposed classification method with other classification method.

Our problem space of 72 basketball games and on the order of 100 two-dimensional end-points per game were computed quickly (on the order of seconds) using k-means. In our framework, k-means was used to cluster trajectory end-points into areas where a shot was taken or a turnover made. Clustering of trajectories (using DTW and hierarchical clustering) was then performed on trajectories terminating in a given area. This was clarified in the methods:

(line 105) Our framework comprises: (i) clustering of trajectory endpoints into areas (where a shot is taken or possession lost, Section 3.2), (ii) DTW to cluster trajectories terminating in each area (Section 3.1 and Section 3.2), and (iii) analysis of play outcomes for each cluster (Section 3.5). In addition, methods for validation (Section 3.3) and generating a representative trajectory from a cluster (Section 3.4) are also provided. This framework demonstrates how the combination of machine learning and statistical methods can be used in novel ways to address a practical problem. 

We have also taken on board the reviewer’s point and have added an investigation of alternate clustering techniques as future work:

(line 385) Potential future work includes optimisation of the clustering process to better account for large numbers of clusters with a small number of trajectories. The current framework utilises a basic implementation of both DTW and k-means clustering; more sophisticated approaches, such as PrunedDTW, SparseDTW, FastDTW, and MultiscaleDTW, could potentially enable faster computation and better clustering results (31). Alternative techniques for clustering and even mixture modelling could be investigated, especially if integrating clustering with trajectory alignment. A higher resolution dataset with better sampling consistency may also assist in the DTW and clustering process; further investigation could include the breakdown of trajectories into discrete actions as part of a vocabulary of basketball actions (15). Finally, extension of the work to include both ball and player trajectories could provide further detail and insight to support coaches and athletes. 

We do not understand the reviewer’s comment that k-means is a “lazy classifier”; we would like to know what is meant here and would be happy to address or expand on this if needed.

To our knowledge, a “lazy classifier” refers to algorithms like kNN which store training data and do not do any work until testing time (Vijayarani and Muthulakshmi, 2013). K-means is a clustering algorithm, and we use it for this purpose.

Vijayarani, S., & Muthulakshmi, M. (2013). Comparative analysis of bayes and lazy classification algorithms. International Journal of Advanced Research in Computer and Communication Engineering, 2(8), 3118-3124.

 In line 143-144, the authors wrote: “parameters such as boundary condition, continuity condition, monotonic condition and warping window condition were left the same during the analysis.” More detailed explanation on these parameters are needed to make this research reproducible.

These parameters are mathematical and algorithmic conditions that determines the alignment behavior process. In this paper we have utilized a pre-coded Dynamic Time Warping package that inputs generic Time series data and computes DTW distance measures and clusters. The parameters are in built to the package and were left as default. The line has been removed to avoid confusion.

1 http://sites.labic.icmc.usp.br/prunedDTW/

2 https://arxiv.org/abs/1201.2969

3 https://cs.fit.edu/~pkc/papers/tdm04.pdf

4 https://www.researchgate.net/publication/334413562_Iterative_Multiscale_Dynamic_Time_Warping_IMs- DTW_A_tool_for_rainfall_time_series_comparison

 In this research, the authors used a dataset of 72 women’s international basketball games ranging from the 2014 FIB world Championship to the 2016 Rio Olympic Games. Was there any specific justification in choosing this dataset? In fact, there are other datasets available in the internet, such as a dataset provided by Rajiv Shah5.

Our work was motivated primarily to provide decision support, via the Australian Institute of Sport, to the national women’s Olympic basketball team. Hence, this dataset was selected to address our collaborator / end-user’s problem.

 More detailed discussions are needed, for instance:

 How does this research contribute in the field of machine learning, especially compared with previous works of time series analysis in sports data? Comparison with previous works is needed to see that the proposed method is better than current state-of-the-arts in the field.

 What are limitations of this research? What is the trade-off of implementing DTW and K-Means clustering?

As per our response to the first comment about the contribution of the paper, our contribution is the development of a framework, comprising existing methods, to relate clusters of plays, as represented by ball movement trajectories, to play outcomes. Hence, our contribution is an applied one, specifically to basketball trajectory and play outcomes analysis. To that end, we have reviewed and contextualised our contribution with respect to the literature in this area, clarified in both the Introduction and Discussion. In the Discussion we have also included limitations and potential future research directions.

Introduction

(line 63) Dynamic Time Warping (DTW) presents an approach to align features, such as a kick-out pass or a baseline drive, in space and time between trajectories of possibly different lengths. Aligning or registering features can be useful as two otherwise identical trajectories where one is slightly offset in space and/or time will have a large mathematical dissimilarity (e.g. Euclidean distance) even though they are similar in practice (6). A registration method such as DTW can simultaneously align features and compute a dissimilarity measure which can then be used to better cluster trajectories. DTW has been used widely in a range of applications due to its computational efficiency and utility for feature alignment, such as the analysis of audio files (7) and in facial recognition as an accurate predictor for classification algorithms (8). DTW classification has also been used to monitor production cycle rates for construction with up to 92% accuracy (9). In addition, the DTW Barycentre Averaging (DBA) algorithm, which is based on DTW, can be used to generate an average trajectory for each cluster (10). DTW has been used to for the purposes of curve registration when classifying events, such as the type of shot being made, using wearable sensor data or video data, with examples in tennis, squats and basketball (11) (12) (13) (14). There are comparatively fewer works in the analysis of basketball plays. Chen et al. (2015) (16) present an approach based on DTW registration of player trajectories and clustering of similar plays, representing similar plays with Gaussian mixture regression. However, their focus was on unsupervised clustering and they did not explicitly relate clusters to play outcomes. Another method that incorporated DTW was the extension of latent Dirichlet allocation to create a vocabulary of actions based on player tracking that combine to form a play (15). This particular work largely ignores temporal effects and focuses on spatial characteristics of actions that make up a play.

Discussion: Conclusion

Our DTW based framework enables analysis and interpretation of complex trajectories over many games and teams, and the generation of key summary metrics of play outcomes. Comparison of trajectory clusters within and between countries can assist in identifying key patterns to support coaching and preparation for games. A key component of this is the ability to highlight key plays for targeted video review from a large database of game footage, assisting end users with limited time resources. A common methodology used in current and previous research utilises Euclidean distances (3)(4)(5) which does not consider data alignment and will miss out similar plays when analysed under unsupervised conditions. Existing works that used DTW in basketball tended to focus on player movements, and did not explicitly relate trajectories to play outcomes with a focus on both spatial and temporal characteristic of the trajectories (16) (15). Our framework contributes a practical means to cluster spatio-temporal trajectories and relate them to play outcomes, using ball movement data. However, the use of k-means to cluster trajectory endpoints is limited in that it cannot take into account boundary conditions associated with court boundaries. In addition, there is immense diversity in trajectories and determining how many clusters to use is somewhat subjective; potentially, a combination of using a vocabulary of discrete actions (15) and a Diricihlet Process Mixture Model (30) could better cluster trajectories and inherently finds the optimal number of clusters. 

*** EOF ***

Reviewer #2: I liked how this paper took a slightly different data science approach and applied it to get some useful sports analytic insights.

It's hard for me to believe that this is the first application of DTW to sports trajectories. Also, citing some other direct work like this would be good: http://www.lukebornn.com/papers/miller_ssac_2017.pdf So just ensuring that previous related work is acknowledged would be valuable. But even if DTW has been done before for trajectories, this paper did focus on womens basketball games.

Thank you for raising this. We have incorporated that paper and some others into the Introduction, and clarified (in the Abstract, Introduction, Methods and Discussion) that the contribution of our work as a framework, comprising existing methods, to relate clusters of plays, as represented by ball movement trajectories, to play outcomes. 

Introduction: 

(line 90) In this paper, we propose a framework that relates play outcomes to automatically learned clusters of plays, vis a vis ball trajectories.

Introduction: 

(line 63) Dynamic Time Warping (DTW) presents an approach to align features, such as a kick-out pass or a baseline drive, in space and time between trajectories of possibly different lengths. Aligning or registering features can be useful as two otherwise identical trajectories where one is slightly offset in space and/or time will have a large mathematical dissimilarity (e.g. Euclidean distance) even though they are similar in practice (6). A registration method such as DTW can simultaneously align features and compute a dissimilarity measure which can then be used to better cluster trajectories. DTW has been used widely in a range of applications due to its computational efficiency and utility for feature alignment, such as the analysis of audio files (7) and in facial recognition as an accurate predictor for classification algorithms (8). DTW classification has also been used to monitor production cycle rates for construction with up to 92% accuracy (9). In addition, the DTW Barycentre Averaging (DBA) algorithm, which is based on DTW, can be used to generate an average trajectory for each cluster (10). DTW has been used to for the purposes of curve registration when classifying events, such as the type of shot being made, using wearable sensor data or video data, with examples in tennis, squats and basketball (11) (12) (13) (14). There are comparatively fewer works in the analysis of basketball plays. Chen et al. (2015) (16) present an approach based on DTW registration of player trajectories and clustering of similar plays, representing similar plays with Gaussian mixture regression. However, their focus was on unsupervised clustering and they did not explicitly relate clusters to play outcomes. Another method that incorporated DTW was the extension of latent Dirichlet allocation to create a vocabulary of actions based on player tracking that combine to form a play (15). This particular work largely ignores temporal effects and focuses on spatial characteristics of actions that make up a play.

I appreciate the authors taking time to visualize DTW and help everyone understand how the technique works. I like the detailed background and the application to the games. I could follow how DTW work and the resulting insights. Really great job sharing lots of detail.

Thank you.

For the plays, could you explain why you focused on the full court plays? I have seen previous work that would just focus on trajectories after the ball passed half court. No need to change the analysis, just explaining your decision would be useful.

We focused on full court plays in part to capture fast breaks and the higher scoring rate associated with them, and also the importance of the first 8 seconds of play in affecting the outcome of the play. We have clarified this in the Methods:

(line 141) We propose the application of DTW to three dimensional trajectories (x,y,t) to capture not just the pathway taken by the ball (dimensions in x and y), but also the speed at which it moved along the trajectory (represented as time t). This is critical as the speed of execution of a play is just as important as the play (i.e. ball movements) itself in affecting scoring outcomes (17). The full trajectory of a play was used to identify the variability and impact of temporal-based elements and was noted by industry partner the first 8 seconds of a play to be crucial in scoring (18).

I like that the paper doesn't try to extend itself too far, but is a solid piece of work on applying DTW to basketball. 

Thank you.

5 http://projects.rajivshah.com/blog/2016/04/02/sportvu_analysis/

---

## [Decision Letter · Decision Letter 1]

28 Jul 2022

Classifying ball trajectories in invasion sports using Dynamic Time Warping: a basketball case study

PONE-D-21-32152R1

Dear Dr. Wu,

We’re pleased to inform you that your manuscript has been judged scientifically suitable for publication and will be formally accepted for publication once it meets all outstanding technical requirements.

Kind regards,

Jean-Christophe Nebel, Ph.D

Academic Editor

PLOS ONE

Reviewers' comments:

Reviewer's Responses to Questions

**Comments to the Author**

1. If the authors have adequately addressed your comments raised in a previous round of review and you feel that this manuscript is now acceptable for publication, you may indicate that here to bypass the “Comments to the Author” section, enter your conflict of interest statement in the “Confidential to Editor” section, and submit your "Accept" recommendation.

Reviewer #1: All comments have been addressed

Reviewer #2: All comments have been addressed

2. Is the manuscript technically sound, and do the data support the conclusions?

Reviewer #1: Partly

Reviewer #2: Yes

3. Has the statistical analysis been performed appropriately and rigorously? 

Reviewer #1: Yes

Reviewer #2: Yes

4. Have the authors made all data underlying the findings in their manuscript fully available?

Reviewer #1: Yes

Reviewer #2: Yes

5. Is the manuscript presented in an intelligible fashion and written in standard English?

Reviewer #1: Yes

Reviewer #2: Yes

6. Review Comments to the Author

Reviewer #1: All comments have been addressed by authors. As for DTW parameters that were pre-coded in the DTW package, please clarify in the manuscript that those parameters were left as default.

Reviewer #2: I found the revisions to the manuscript acceptable. I appreciate the authors diligence in incorporating these comments.

7. PLOS authors have the option to publish the peer review history of their article (what does this mean?). If published, this will include your full peer review and any attached files.

Reviewer #1: No

Reviewer #2: **Yes: **Rajiv Shah

---

## [Editor Report · Acceptance letter]

1 Aug 2022

PONE-D-21-32152R1 

Classifying ball trajectories in invasion sports using Dynamic Time Warping: a basketball case study 

Dear Dr. Wu:

I'm pleased to inform you that your manuscript has been deemed suitable for publication in PLOS ONE. Congratulations! Your manuscript is now with our production department. 

Kind regards, 

on behalf of

Prof. Jean-Christophe Nebel 

Academic Editor

PLOS ONE